# Faster DBSCAN via subsampled similarity queries

**Heinrich Jiang***
Google Research
heinrichj@google.com

**Jennifer Jang***
Waymo
jangj@waymo.com

**Jakub Łącki**
Google Research
jlacki@google.com

## Abstract

DBSCAN is a popular density-based clustering algorithm. It computes the $\epsilon$-neighborhood graph of a dataset and uses the connected components of the high-degree nodes to decide the clusters. However, the full neighborhood graph may be too costly to compute with a worst-case complexity of $O(n^2)$. In this paper, we propose a simple variant called SNG-DBSCAN, which clusters based on a subsampled $\epsilon$-neighborhood graph, only requires access to similarity queries for pairs of points and in particular avoids any complex data structures which need the embeddings of the data points themselves. The runtime of the procedure is $O(sn^2)$, where $s$ is the sampling rate. We show under some natural theoretical assumptions that $s \approx \log n/n$ is sufficient for statistical cluster recovery guarantees leading to an $O(n \log n)$ complexity. We provide an extensive experimental analysis showing that on large datasets, one can subsample as little as $0.1\%$ of the neighborhood graph, leading to as much as over 200x speedup and 250x reduction in RAM consumption compared to scikit-learn's implementation of DBSCAN, while still maintaining competitive clustering performance.

## 1   Introduction

DBSCAN [13] is a popular density-based clustering algorithm which has had a wide impact on machine learning and data mining. Recent applications include superpixel segmentation [40], object tracking and detection in self-driving [51, 19], wireless networks [12, 53], GPS [10, 36], social network analysis [29, 55], urban planning [14, 38], and medical imaging [46, 3]. The clusters that DBSCAN discovers are based on the connected components of the neighborhood graph of the data points of sufficiently high density (i.e. those with a sufficiently high number of data points in their neighborhood), where the neighborhood radius and the density threshold are the hyperparameters.

One of the main differences of density-based clustering algorithms such as DBSCAN compared to popular objective-based approaches such as k-means [2] and spectral clustering [50] is that density-based algorithms are non-parametric. As a result, DBSCAN makes very few assumptions on the data, automatically finds the number of clusters, and allows clusters to be of arbitrary shape and size [13]. However, one of the drawbacks is that it has a worst-case quadratic runtime [16]. With the continued growth of modern datasets in both size and richness, non-parametric unsupervised procedures are becoming ever more important in understanding such datasets. Thus, there is a critical need to establish more efficient and scalable versions of these algorithms.

The computation of DBSCAN can be broken up into two steps. The first is computing the $\epsilon$-neighborhood graph of the data points, where the $\epsilon$-neighborhood graph is defined with data points as vertices and edges between pairs of points that are distance at most $\epsilon$ apart. The second is processing the neighborhood graph to extract the clusters. The first step has worst-case quadratic complexity simply due to the fact that each data point may have of order linear number of points in its $\epsilon$-neighborhood for sufficiently high $\epsilon$. However, even if the $\epsilon$-neighborhood graph does not

---

have such an order of edges, computing this graph remains costly: for each data point, we must query for neighbors in its $\epsilon$-neighborhood, which is worst-case linear time for each point. There has been much work done in using space-partitioning data structures such as KD-Trees [4] to improve neighborhood queries, but these methods still run in linear time in the worst-case. Approximate methods (e.g. [23, 9]) answer queries in sub-linear time, but such methods come with few guarantees. The second step is processing the neighborhood graph to extract the clusters, which consists in finding the connected components of the subgraph induced by nodes with degree above a certain threshold (i.e. the MinPts hyperparameter in the original DBSCAN [13]). This step is linear in the number of edges in the $\epsilon$-neighborhood graph.

Our proposal is based on a simple but powerful insight: the full $\epsilon$-neighborhood graph may not be necessary to extract the desired clustering. We show that we can subsample the edges of the neighborhood graph while still preserving the connected components of the core-points (the high density points) on which DBSCAN's clusters are based.

To analyze this idea, we assume that the points are sampled from a distribution defined by a density function satisfying certain standard [26, 44] conditions (e.g., cluster density is sufficiently high, and clusters do not become arbitrarily thin). Such an assumption is natural because DBSCAN recovers the high-density regions as clusters [13, 26]. Under this assumption we show that the minimum cut of the $\epsilon$-neighborhood graph is as large as $\Omega(n)$, where $n$ is the number of datapoints. This, combined with a sampling lemma by Karger [28], implies that we can sample as little as $O(\log n/n)$ of the edges uniformly while preserving the connected components of the $\epsilon$-neighborhood graph *exactly*. - Our algorithm, SNG-DBSCAN, proceeds by constructing and processing this subsampled $\epsilon$-neighborhood graph all in $O(n \log n)$ time. Moreover, our procedure only requires access to $O(n \log n)$ similarity queries for random pairs of points (adding an edge between pairs if they are at most $\epsilon$ apart). Thus, unlike most implementations of DBSCAN which take advantage of space-partitioning data structures, we don't require the embeddings of the datapoints themselves. In particular, our method is compatible with arbitrary similarity functions instead of being restricted to a handful of distance metrics such as the Euclidean.

We provide an extensive empirical analysis showing that SNG-DBSCAN is effective on real datasets. We show on large datasets (on the order of a million datapoints) that we can subsample as little as $0.1\%$ of the neighborhood graph and attain competitive performance to sci-kit learn's implementation of DBSCAN while consuming far fewer resources – as much as 200x speedup and 250x less RAM consumption on cloud machines with up to 750GB of RAM. In fact, for larger settings of $\epsilon$ on these datasets, DBSCAN fails to run at all due to insufficient RAM. We also show that our method is effective even on smaller datasets. Sampling between $1\%$ to $30\%$ of the edges depending on the dataset, SNG-DBSCAN shows a nice improvement in runtime while still maintaining competitive clustering performance.

## 2  Related Work

There is a large body of work on making DBSCAN more scalable. Due to space we can only mention some of these works here. The first approach is to more efficiently perform the nearest neighbor queries that DBSCAN uses when constructing the neighborhood graph either explicitly or implicitly [21, 31]. However, while these methods do speed up the computation of the neighborhood graph, they may not save memory costs overall as the number of edges remains similar. Our method of subsampling the neighborhood graph brings both memory and computational savings.

A natural idea for speeding up the construction of the nearest neighbors graph is to compute it approximately, for example by using locality-sensitive hashing (LSH) [24] and thus improving the overall running time [41, 54]. At the same time, since the resulting nearest neighbor graph is incomplete, the current state-of-the-art LSH-based methods lack any guarantees on the quality of the solutions they produce. This is in sharp contrast with SNG-DBSCAN, which, under certain assumptions, gives exactly the same result as DBSCAN.

Another approach is to first find a set of "leader" points that preserve the structure of the original dataset and cluster those "leader" points first. Then the remaining points are clustered to these "leader points" [17, 49]. Liu [32] modified DBSCAN by selecting clustering seeds among unlabeled core points to reduce computation time in regions that have already been clustered. Other heuristics include [37, 30]. More recently, Jang and Jiang [25] pointed out that it's not necessary to compute

the density estimates for each of the data points and presented a method that chooses a subsample of the data using $k$-centers to save runtime on density computations. These approaches all reduce the number of data points on which we need to perform the expensive neighborhood graph computation. SNG-DBSCAN preserves all of the data points but subsamples the edges instead.

There are also a number of approaches based on leveraging parallel computing [7, 37, 18], which includes MapReduce based approaches [15, 20, 8, 35]. Then there are also distributed approaches to DBSCAN where data is partitioned across different locations, and there may be communication cost constraints [34, 33]. Andrade et al. [1] provides a GPU implementation. In this paper, we assume a single processor, although our method can be implemented in parallel which can be a future research direction.

## 3  Algorithm

---

**Algorithm 1** Subsampled Neighborhood Graph DBSCAN (SNG-DBSCAN)

---

**Inputs:** $X_{[n]}$, sampling rate $s$, $\epsilon$, MinPts
Initialize graph $G = (X_{[n]}, \varnothing)$.
For each $x \in X_{[n]}$, sample $\lceil sn \rceil$ examples from $X_{[n]}$, $x_{i_1}, ..., x_{i_{\lceil sn \rceil}}$ and add edge $(x, x_{i_j})$ to $G$ if $|x - x_{i_j}| \leqslant \epsilon$ for $j \in [\lceil sn \rceil]$.
Let $\mathcal{K} := \{K_1, ..., K_\ell\}$ be the connected components of the subgraph of $G$ induced by vertices of degree at least MinPts.
Initialize $C_i = \varnothing$ for $i \in [\ell]$ and define $\mathcal{C} := \{C_1, ..., C_\ell\}$. For each $x \in X_{[n]}$, add $x$ to $C_i$ if $x \in K_i$. Otherwise, if $x$ is connected to some point in $K_i$, add $x$ to $C_i$ (if multiple such $i \in [\ell]$ exist, choose one arbitrarily).
**return** $\mathcal{C}$.

---

We now introduce our algorithm SNG-DBSCAN (Algorithm 1). It proceeds by first constructing the sampled $\epsilon$-neighborhood graph given a sampling rate $s$. We do this by initializing a graph whose vertices are the data points, sampling $s$ fraction of all pairs of data points, and adding corresponding edges to the graph if points are less than $\epsilon$ apart. Compared to DBSCAN, the latter computes the full $\epsilon$-neighborhood graph, typically using space-partitioning data structures such as kd-trees, while SNG-DBSCAN can be seen as using a sampled version of brute-force (which looks at each pair of points to compute the graph). Despite not leveraging space-partitioning data structures, we show in the experiments that SNG-DBSCAN can still be much more scalable in both runtime and memory than DBSCAN.

The remaining stage of processing the graph is the same as in DBSCAN, where we find the core-points (points with degree at least degree MinPts), compute the connected components induced by the core-points, and then cluster the border-points (those within $\epsilon$ of a connected component). The remaining points are not clustered and are referred to as noise-points or outliers. The entire procedure runs in $O(sn^2)$ time. We will show in the theory that $s$ can be of order $O(\log n / n)$ while still maintaining statistical consistency guarantees of recovering the

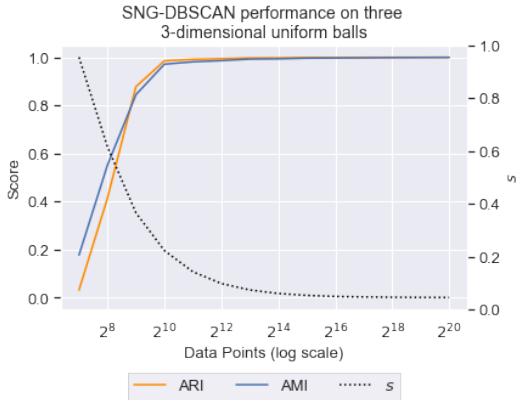

Figure 1: **SNG-DBSCAN on simulated uniform balls**. Data is generated as a uniform mixture of 3 balls in $\mathbb{R}^3$, where each ball is a *true* cluster. The $x$-axis is the number of examples $n$ on a log scale. We set $s = \lceil 20 \cdot \log n / n \rceil$ and show the performance of SNG-DBSCAN under two clustering metrics Adjusted Rand Index (ARI) and Adjusted Mutual Information (AMI). We see that indeed, sampling $s \approx \log n / n$ is sufficient to recover the right clustering for sufficiently large $n$, as the theory suggests. Clustering quality is low at the beginning because the sample size was insufficient to learn the clusters.

*true* clusters under certain density assumptions. Under this setting, the procedure runs in $O(n \log n)$ time, and we show on simulations in Figure 1 that such a setting for $s$ is sufficient.

# 4 Theoretical Analysis

For the analysis, we assume that we draw $n$ i.i.d. samples $X_{[n]} := \{x_1, ..., x_n\}$ from a distribution $\mathcal{P}$. The goal is to show that the sampled version of DBSCAN (Algorithm 1) can recover the true clusters with statistical guarantees, where the true clusters are the connected components of a particular upper-level set of the underlying density function, as it is known that DBSCAN recovers such connected components of a level set [26, 52, 43].

We show results in two situations: (1) when the clusters are well-separated and (2) results for recovering the clusters of a particular level of the density under more general assumptions.

## 4.1 Recovering Well-Separated Clusters

We make the following Assumption 1 on $\mathcal{P}$. The assumption is three-fold. The first part ensures that the true clusters are of sufficiently high density level and the noise areas are of sufficiently low density. The second part ensures that the true clusters are pairwise separated by a sufficiently wide distance and that they are also separated away from the noise regions; such an assumption is common in analyses of cluster trees (e.g. [5, 6, 26]). Finally, the last part ensures that the clusters don't become arbitrarily thin anywhere. Otherwise it would be difficult to show that the entire cluster will be recovered as one connected component. This has been used in other analyses of level-set estimation [42].

**Assumption 1.** *Data is drawn from distribution $\mathcal{P}$ on $\mathbb{R}^D$ with density function $p$. There exists $R_s, R_0, \rho, \lambda_C, \lambda_N > 0$ and compact connected sets $C_1, ..., C_\ell \subset \mathbb{R}^D$ and subset $N \subset \mathbb{R}^D$ such that*

- *$p(x) \geqslant \lambda_C$ for all $x \in C_i, i \in [\ell]$ (i.e. clusters are high-density), $p(x) \leqslant \lambda_N$ for all $x \in N$ (i.e. outlier regions are low-density), $p(x) = 0$ everywhere else, and that $\rho \cdot \lambda_C > \lambda_N$ (the cluster density is sufficiently higher than noise density).*

- *$\min_{x \in C_i, x' \in C_j} |x - x'| \geqslant R_s$ for $i \neq j$ and $i, j \in [\ell]$ (i.e clusters are separated) and $\min_{x \in C_i, x' \in N} |x - x'| \geqslant R_s$ for $i \in [\ell]$ (i.e. outlier regions are away from the clusters).*

- *For all $0 < r < R_0$, $x \in C_i$ and $i \in [\ell]$ we have $Volume(B(x, r) \cap C_i) \geqslant \rho \cdot v_D \cdot r^D$ where $B(x, r) := \{x' : |x - x'| \leqslant r\}$ and $v_D$ is the volume of a unit $D$-dimensional ball. (i.e. clusters don't become arbitrarily thin).*

The analysis proceeds in three steps:

1. We give a lower bound on the min-cut of the subgraph of the $\epsilon$-neighborhood graph corresponding to each cluster. This will be useful later as it determines the sampling rate we can use while still ensure these subgraphs remain connected. (Lemma 1)

2. We use standard concentration inequalities to show that if MinPts is set appropriately, we will with high probability determine which samples belong to clusters and which ones don't in the sampled graph. (Lemma 2)

3. We then combine these two results to give a precise bound on the sampling rate $s$ and sample complexity $n$ to show that Algorithm 1 properly identifies the clusters. (Theorem 1)

We now give the lower bound on the min-cut of the subgraph of the $\epsilon$-neighborhood graph corresponding to each cluster. This will be useful later as it determines the sampling rate we can use while still ensuring these subgraphs remain connected. As a reminder, for a graph $G = (V, E)$, the size of the cut-set of $S \subseteq V$ is defined as $\text{Cut}_G(S, V \backslash S) := |\{(p, q) \in E : p \in S, q \in V \backslash S\}|$ and the size of the min-cut of $G$ is the smallest proper cut-set size: $\text{MinCut}(G) := \min_{S \subset V, S \neq \varnothing, S \neq V} \text{Cut}_G(S, V \backslash S)$.

**Lemma 1** (Lower bound on min-cut of $\epsilon$-neighborhood graph of core-points). *Suppose that Assumption 1 holds and $\epsilon < R_0$. Let $\gamma > 0$. Then there exists a constant $C_{\delta, D, p}$ depending only on $D$ and $p$ such that the following holds for $n \geqslant C_{D,p} \cdot \log(2/\delta) \cdot \left(\frac{1}{\epsilon}\right)^D \cdot \log^{1+\gamma}\left(\frac{1}{\epsilon}\right)$. Let $G_{n,\epsilon}$ be the*

$\epsilon$-neighborhood graph of $X_{[n]}$. Let $G_{n,\epsilon}(i)$ be the subgraph of $G_{n,\epsilon}$ with nodes in $C_i$. Then with probability at least $1 - \delta$, for each $i \in [\ell]$, we have that

$$MinCut(G_{n,\epsilon}(i)) \geqslant \frac{1}{4} \cdot \lambda_C \cdot \rho \cdot v_D \cdot \epsilon^D \cdot n.$$

We next show that if MinPts is set appropriately, we will with high probability determine which samples belong to clusters and which ones don't in the sampled graph.

**Lemma 2.** *Suppose that Assumption 1 holds and $\epsilon < \min\{R_0, R_S\}$. Let $\delta, \gamma > 0$. There exists a universal constant $C$ such that the following holds. Let the sampling rate be $s$ and suppose*

$$\lambda_N \cdot v_D \cdot \epsilon^D < \frac{minPts}{sn} < \rho \cdot \lambda_C \cdot v_D \cdot \epsilon^D,$$

*and $sn \geqslant \frac{C}{\Delta^2} \cdot \log^{1+\gamma}\left(\frac{1}{\Delta}\right) \cdot \log(1/\delta)$, where $\Delta := \min\{\frac{minPts}{sn} - \lambda_N \cdot v_D \cdot \epsilon^D, \rho \cdot \lambda_C \cdot v_D \cdot \epsilon^D - \frac{minPts}{sn}\}$. Then with probability at least $1 - \delta$, all samples in $C_i$ for some $i \in [\ell]$ are identified as core-points and the rest are noise points.*

The next result shows a rate at which we can sample a graph while still have it be connected, which depends on the min-cut and the size of the graph. It follows from classical results in graph theory that cut sizes remain preserved under sampling [28].

**Lemma 3.** *There exists universal constant $C$ such that the following holds. Let $G$ be a graph with min-cut $m$ and $0 < \delta < 1$. If*

$$s \geqslant \frac{C \cdot (\log(1/\delta) + \log(n))}{m},$$

*then with probability at least $1 - \delta$, the graph $G_s$ obtained by sampling each edge of $G$ with probability $s$ is connected.*

We now give the final result, which follows from combining Lemmas 1, 2, and 3.

**Theorem 1.** *Suppose that Assumption 1 holds and $\epsilon < \min\{R_0, R_S\}$. Let $\delta, \gamma > 0$. There exist universal constants $C_1, C_2$ and constant $C_{D,p}$ depending only on $D$ and $p$ such that the following holds. Suppose*

$$\lambda_N \cdot v_D \cdot \epsilon^D < \frac{minPts}{sn} < \rho \cdot \lambda_C \cdot v_D \cdot \epsilon^D, \qquad s \geqslant \frac{C_1 \cdot (\log(1/\delta) + \log n)}{\lambda_C \cdot \rho \cdot v_D \cdot \epsilon^D \cdot n},$$

*and*

$$n \geqslant \max\left\{ C_{D,p} \cdot \log(2/\delta) \cdot \left(\frac{1}{\epsilon}\right)^D \cdot \log^{1+\gamma}\left(\frac{1}{\epsilon}\right), \frac{1}{s}\frac{C_2}{\Delta^2} \cdot \log^{1+\gamma}\left(\frac{1}{\Delta}\right) \cdot \log(1/\delta) \right\},$$

*where $\Delta := \min\{\frac{minPts}{sn} - \lambda_N \cdot v_D \cdot \epsilon^D, \rho \cdot \lambda_C \cdot v_D \cdot \epsilon^D - \frac{minPts}{sn}\}$.*

*Then with probability at least $1 - 3\delta$, Algorithm 1 returns (up to permutation) the clusters $\{C_1 \cap X_{[n]}, C_2 \cap X_{[n]}, ..., C_\ell \cap X_{[n]}\}$.*

**Remark 1.** *As a consequence, we can take $s = \Theta(\log n/\epsilon^D n)$, and, for $n$ sufficiently large, the clusters are recovered with high probability, leading to a computational complexity of $O(\frac{1}{\epsilon^D} \cdot n \log n)$ for Algorithm 1.*

### 4.2 Recovering Clusters of a Particular Level of the Density

We show level-set estimation rates for estimating a particular level $\lambda$ (i.e. $L_f(\lambda) := \{x \in \mathcal{X} : f(x) \geqslant \lambda\}$) given that hyperparameters of SNG-DBSCAN are set appropriately depending on density $f$, $s$, $\lambda$ and $n$ under the following assumption which characterizes the regularity of the density function around the boundaries of the level-set. This assumption is standard amongst analyses in level-set estimation e.g. [26, 42].

**Assumption 2.** *$f$ is a uniformly continuous density on compact set $\mathcal{X} \subseteq \mathbb{R}^D$. There exists $\beta, \check{C}, \hat{C}, r_c > 0$ such that the following holds for all $x \in B(L_f(\lambda), r_c)\backslash L_f(\lambda)$: $\check{C} \cdot d(x, L_f(\lambda))^\beta \leqslant \lambda - f(x) \leqslant \hat{C} \cdot d(x, L_f(\lambda))^\beta$, where $d(x, A) := \inf_{x' \in A} |x - x'|$ and $B(A, r) := \{x : d(x, A) \leqslant r\}$.*

**Theorem 2.** *Suppose that Assumption 2 holds, $0 < \delta < 1$, and $\epsilon$ and MinPts satisfies the following:*

$$\epsilon^\beta = \frac{1}{\widehat{C}} \left( \lambda - \frac{MinPts}{v_D \cdot \epsilon^D \cdot s \cdot n} - \sqrt{\frac{\log(4n) + \log(1/\delta)}{s \cdot n}} \right).$$

*Let $\widehat{L_f(\lambda)}$ be the union of all the clusters returned by SNG-DBSCAN. Then, for $n$ sufficiently large depending on $f, \epsilon, MinPts$, the following holds with probability at least $1 - \delta$:*

$$d_{Haus}(\widehat{L_f(\lambda)}, L_f(\lambda)) \leqslant C \cdot \left( \left( \frac{\log(4n) + \log(1/\delta)}{s \cdot n} \right)^{1/2\beta} + \epsilon \right),$$

*where $C$ is some constant depending on $f$ and $d_{Haus}$ is Hausdorff distance.*

Given these results, they can be extended them to obtain clustering results with the same convergence rates (i.e. showing that SNG-DBSCAN recovers the connected components of the level-set individually) in a similar way as shown in the previous result.

## 5   Experiments

**Datasets and hyperparameter settings**: We compare the performance of SNG-DBSCAN against DBSCAN on 5 large (~1,000,000+ datapoints) and 12 smaller (~100 to ~100,000 datapoints) datasets from UCI [11] and OpenML [47]. Details about each large dataset shown in the main text are summarized in Figure 2, and the rest, including all hyperparameter settings, can be found in the Appendix. Due to space constraints, we couldn't show the results for all of the datasets in the main text. The settings of MinPts and range of $\epsilon$ that we ran SNG-DBSCAN and DBSCAN on, as well as how $s$ was chosen, are shown in the Appendix. For simplicity we fixed MinPts and only tuned $\epsilon$ which is the more essential hyperparameter. We compare our implementation of SNG-DBSCAN to that of sci-kit learn's DBSCAN [39], both of which are implemented with Cython, which allows the code to have a Python API while the expensive computations are done in a C/C++ backend.

**Clustering evaluation**: To score cluster quality, we use two popular clustering scores: the Adjusted Rand Index (ARI) [22] and Adjusted Mutual Information (AMI) [48] scores. ARI is a measure of the fraction of pairs of points that are correctly clustered to the same or different clusters. AMI is a measure of the cross-entropy of the clusters and the ground truth. Both are normalized and adjusted for chance, so that the perfect clustering receives a score of 1 and a random one receives a score of 0. The datasets we use are classification datasets where we cluster on the features and use the labels to evaluate our algorithm's clustering performance. It is standard practice to evaluate performance using the labels as a proxy for the ground truth [27, 25].

### 5.1   Performance on Large Datasets

We now discuss the performance on the large datasets, which we ran on a cloud compute environment. Due to computational costs, we only ran DBSCAN once for each $\epsilon$ setting. We ran SNG-DBSCAN 10 times for each $\epsilon$ setting and averaged the clustering scores and runtimes. The results are summarized in Figure 3, which reports the highest clustering scores for the respective algorithms along with the runtime and RAM usage required to achieve these scores. We also plot the clustering performance and runtime/RAM usage across different settings of $\epsilon$ in Figure 4.

|  | $n$ | $D$ | $c$ | $s$ | mPts |
|---|---|---|---|---|---|
| Australian | 1,000,000 | 14 | 2 | 0.001 | 10 |
| Still [45] | 949,983 | 3 | 6 | 0.001 | 10 |
| Watch [45] | 3,205,431 | 3 | 7 | 0.01 | 10 |
| Satimage | 1,000,000 | 36 | 6 | 0.02 | 10 |
| Phone [45] | 1,000,000 | 3 | 7 | 0.001 | 10 |

Figure 2: *Summary of larger datasets and hyperparameter settings used.* Includes dataset size ($n$), number of features ($D$), number of clusters ($c$), and the fraction $s$ of neighborhood edges kept by SNG-DBSCAN. Datasets Phone, Watch, and Still come from the Heterogeneity Activity Recognition dataset. Phone originally had over 13M points, but in order to feasibly run DBSCAN, we used a random 1M sample. For the full chart, including settings for $\epsilon$, see the Appendix.

|  | Base ARI | SNG ARI | Base AMI | SNG AMI |
|---|---|---|---|---|
| Australian | **0.0933** (6h 4m) | **0.0933** (1m 36s) | **0.1168** (4h 28m) | 0.1167 (1m 36s) |
|  | 282 GB | 1.5 GB | 146 GB | 1.5 GB |
| Still | 0.7901 (14m 37s) | **0.7902** (1m 21s) | 0.8356 (14m 37s) | **0.8368** (1m 57s) |
|  | 419 GB | 1.6 GB | 419 GB | 1.5 GB |
| Watch | 0.1360 (27m 2s) | **0.1400** (1h 27m) | 0.1755 (7m 35s) | **0.1851** (1h 29m) |
|  | 518 GB | 8.7 GB | 139 GB | 7.6 GB |
| Satimage | 0.0975 (1d 3h) | **0.0981** (33m 22s) | 0.1019 (1d 3h) | **0.1058** (33m 22s) |
|  | 11 GB | 2.1 GB | 11 GB | 2.1 GB |
| Phone | 0.1902 (12m 36s) | **0.1923** (59s) | 0.2271 (4m 48s) | **0.2344** (46s) |
|  | 138 GB | 1.7 GB | 32 GB | 1.5 GB |

Figure 3: *Performance on large datasets.* Best Adjusted Rand Index and Adjusted Mutual Information Scores for both DBSCAN and SNG-DBSCAN after $\epsilon$-tuning, shown with their associated runtimes and RAM used. Highest scores for each dataset are bolded. We see that SNG-DBSCAN is competitive against DBSCAN in terms of clustering quality while using much fewer resources. On Australian, we see an over 200x speedup. On Still, we see an over 250x reduction in RAM.

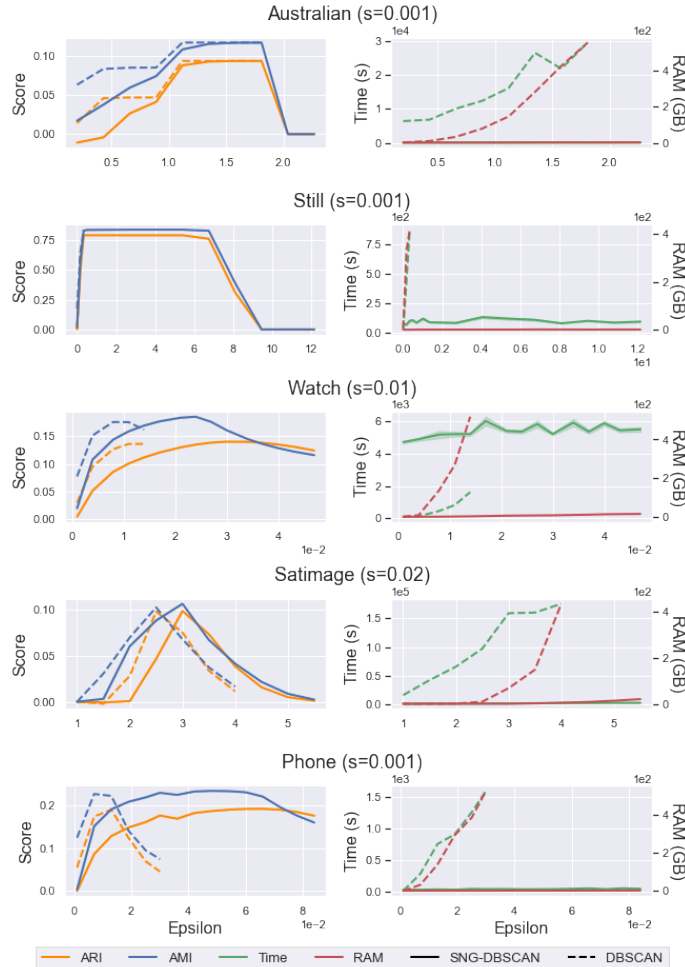

Figure 4: **Large dataset results across** $\epsilon$. We show the performance of SNG-DBSCAN and DB-SCAN on five large datasets. SNG-DBSCAN values are averaged over 10 runs, and 95% confidence intervals (standard errors) are shown. We ran these experiments on a cloud environment and plot the adjusted RAND index score, adjusted mutual information score, runtime, and RAM used across a wide range of $\epsilon$. DBSCAN was run with MinPts $= 10$, and SNG-DBSCAN was run with MinPts $= \max(2, \lfloor 10 \cdot s \rfloor)$ for all datasets. No data is given for DBSCAN for $\epsilon$ settings requiring more than 750GB of RAM as it wasn't possible for DBSCAN to run on these machines.

From Figure 4, we see that DBSCAN often exceeds the 750GB RAM limit on our machines and fails to complete; the amount of memory (as well as runtime) required for DBSCAN escalates quickly as $\epsilon$ increases, suggesting that the size of the $\epsilon$-neighborhood graph grows quickly. Meanwhile, SNG-DBSCAN's memory usage remains reasonable and increases slowly across $\epsilon$. This suggests that SNG-DBSCAN can run on much larger datasets infeasible for DBSCAN, opening the possibilities for applications which may have previously not been possible due to scalability constraints – all the while attaining competitive clustering quality (Figure 3).

Similarly, SNG-DBSCAN shows a significant runtime improvement for almost all datasets and stays relatively constant across epsilon. We note that occasionally (e.g. Watch dataset for small $\epsilon$), SNG-DBSCAN is slower. This is likely because SNG-DBSCAN does not take advantage of space-partitioning data structures such as kd-trees. This is more likely on lower dimensional datasets where space-partitioning data structures tend to perform faster as the number of partitions is exponential in dimension [4]. Conversely, we see the largest speedup on Australian, which is the highest dimension large dataset we evaluated. However, DBSCAN was unable to finish clustering Watch past the first few values of $\epsilon$ due to exceeding memory limits. During preliminary experimentation, but not shown in the results, we tested DBSCAN on the UCI Character Font Images dataset which has dimension 411 and 745,000 datapoints. DBSCAN failed to finish after running on the cloud machine for over 6 days. Meanwhile, SNG-DBSCAN was able to finish with the same $\epsilon$ settings in under 15 mins using the $s = 0.01$ setting. We didn't show the results here because we were unable to obtain clustering scores for DBSCAN.

## 5.2 Performance on smaller datasets

We now show that we don't require large datasets to enjoy the advantages of SNG-DBSCAN. Speedups are attainable for even the smallest datasets without sacrificing clustering quality. Due to space constraints, we provide the summary of the datasets and hyperparameter setting details in the Appendix. In Figure 5, we show performance metrics for some datasets under optimal tuning. The results for the rest of the datasets are in the Appendix, where we also provide charts showing performance across $\epsilon$ and different settings of $s$ to better understand the effect of the sampling rate on cluster quality– there we show that SNG-DBSCAN is stable in the $s$ hyperparameter. Overall, we see that SNG-DBSCAN can give considerable savings in computational costs while remaining competitive in clustering quality on smaller datasets.

|             | Base ARI        | SNG ARI         | Base AMI        | SNG AMI         |
|-------------|-----------------|-----------------|-----------------|-----------------|
| Wearable    | 0.2626 (55s)    | **0.3064** (8.2s) | **0.4788** (26s) | 0.4720 (6.6s)   |
| Iris        | **0.5681** (<0.01s) | **0.5681** (<0.01s) | **0.7316** (<0.01s) | **0.7316** (<0.01s) |
| LIBRAS      | 0.0713 (0.03s)  | **0.0903** (<0.01s) | 0.2711 (0.02s)  | **0.3178** (<0.01s) |
| Page Blocks | 0.1118 (0.38s)  | **0.1134** (0.06s) | **0.0742** (0.49s) | 0.0739 (0.06s)  |
| kc2         | 0.3729 (<0.01s) | **0.3733** (<0.01s) | **0.1772** (<0.01s) | 0.1671 (<0.01s) |
| Faces       | 0.0345 (1.7s)   | **0.0409** (0.16s) | 0.2399 (1.7s)   | **0.2781** (0.15s) |
| Ozone       | 0.0391 (<0.01s) | **0.0494** (<0.01s) | 0.1214 (<0.01s) | **0.1278** (<0.01s) |
| Bank        | 0.1948 (3.4s)   | **0.2265** (0.03s) | 0.0721 (3.3s)   | **0.0858** (0.03s) |

Figure 5: *Clustering performance.* Best Adjusted Rand Index and Adjusted Mutual Information scores for both DBSCAN and SNG-DBSCAN under optimal tuning are given with associated runtimes in parentheses. Highest scores for each dataset are bolded. Full table can be found in the Appendix.

## 5.3 Performance against DBSCAN++

We now compare the performance of SNG-DBSCAN against a recent DBSCAN speedup called DBSCAN++[25], which proceeds by performing $k$-centers to select $\lfloor sn \rfloor$ candidate points, computing densities for these candidate points and using these densities to identify a subset of core-points, and finally clustering the dataset based on the $\epsilon$-neighborhood graph of these core-points. Like our method, DBSCAN++ also has $O(sn^2)$ runtime. SNG-DBSCAN can be seen as subsampling edges while DBSCAN++ subsamples the vertices.

We show comparative results on our large datasets in Figure 6. We ran DBSCAN++ with the same sampling rate $s$ as SNG-DBSCAN. We see that SNG-DBSCAN is competitive and beats DBSCAN++ on 70% of the metrics. Occasionally, DBSCAN++ fails to produce reasonable clusters with the $s$ given, as shown by the low scores for some datasets. We see that SNG-DBSCAN is slower than DBSCAN++ for the low-dimensional datasets (i.e. Phone, Watch, and Still, which are all of dimension 3) while there is a considerable speedup on Australian. Like in the experiments on the large datasets against DBSCAN, this is due to the fact that DBSCAN++, like DBSCAN, leverages space-partitioning data structures such as kd-trees, which are faster in low dimensions. Overall, we see that SNG-DBSCAN is a better choice over DBSCAN++ when using the same sampling rate.

|  | DBSCAN++ ARI | SNG ARI | DBSCAN++ AMI | SNG AMI |
|---|---|---|---|---|
| Australian | **0.1190** (1185s) 1.6 GB | 0.0933 (96s) 1.5 GB | 0.1166 (1178s) 1.6 GB | **0.1167** (96s) 1.5 GB |
| Satimage | 0.0176 (5891s) 2.3 GB | **0.0981** (2002s) 2.1 GB | **0.1699** (4842s) 2.3 GB | 0.1058 (2002s) 2.1 GB |
| Phone | 0.0001 (11s) 1.0 GB | **0.1923** (59s) 1.7 GB | 0.0023 (11s) 1.0 GB | **0.2344** (46s) 1.5 GB |
| Watch | 0.0000 (973s) 1.4 GB | **0.1400** (5230s) 8.7 GB | 0.0025 (1017s) 1.4 GB | **0.1851** (5371s) 7.6 GB |
| Still | 0.7900 (12s) 1.4 GB | **0.7902** (81s) 1.6 GB | **0.8370** (15s) 1.4 GB | 0.8368 (117s) 1.5 GB |

Figure 6: *SNG-DBSCAN against DBSCAN++[25] on large datasets tuned over $\epsilon$.*

|  | DBSCAN++ ARI | SNG ARI | DBSCAN++ AMI | SNG AMI |
|---|---|---|---|---|
| Wearable | 0.2517 | **0.3263** | 0.4366 | **0.5003** |
| Iris | **0.6844** | 0.5687 | **0.7391** | 0.7316 |
| LIBRAS | **0.1747** | 0.0904 | **0.3709** | 0.2985 |
| Page Blocks | 0.0727 | **0.1137** | 0.0586 | **0.0760** |
| kc2 | 0.3621 | **0.3747** | 0.1780 | **0.1792** |
| Faces | **0.0441** | 0.0424 | **0.2912** | 0.2749 |
| Ozone | **0.0627** | 0.0552 | 0.1065 | **0.1444** |
| Bank | **0.2599** | 0.2245 | 0.0874 | **0.0875** |
| Ionosphere | 0.1986 | **0.6359** | 0.2153 | **0.5615** |
| Mozilla | 0.1213 | **0.2791** | 0.1589 | **0.1806** |
| Tokyo | 0.4180 | **0.4467** | 0.2793 | **0.3147** |
| Vehicle | **0.1071** | 0.0971 | **0.1837** | 0.1807 |

Figure 7: *Performance comparison of SNG-DBSCAN and DBSCAN++ (with uniform sampling) with $\epsilon$ tuned appropriately and sampling rate tuned over grid [0.1,0.2,..,0.9] to maximize ARI and AMI clustering scores.*

In Figure 7, we compare DBSCAN++ and SNG-DBSCAN on the smaller datasets under optimal tuning of the sampling rate as well as the other hyperparameters. We find that under optimal tuning, these algorithms have competitive performance against each other.

## 6   Conclusion

Density clustering has had a profound impact on machine learning and data mining; however, it can be very expensive to run on large datasets. We showed that the simple idea of subsampling the neighborhood graph leads to a procedure, which we call SNG-DBSCAN that runs in $O(n \log n)$ time and comes with statistical consistency guarantees. We showed empirically on a variety of real datasets that SNG-DBSCAN can offer a tremendous savings in computational resources while maintaining clustering quality. Future research directions include using adaptive instead of uniform sampling of edges, combining SNG-DBSCAN with other speedup techniques, and paralellizing the procedure for even faster computation.

## Broader Impact

As stated in the introduction, DBSCAN has a wide range of applications within machine learning and data mining. Our contribution is a more efficient variant of DBSCAN. The potential impact of SNG-DBSCAN lies in considerable savings in computational resources and further applications of density clustering which weren't possible before due to scalability constraints.

## Acknowledgments and Disclosure of Funding

Part of the work was done when Jennifer Jang was employed at Uber. Jennifer also received unemployment benefits from the New York State Department of Labor during the time she worked on this paper.

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
