[Supplementary Material]

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

| | 1.6 GB | 1.5 GB | 1.6 GB | 1.5 GB |
| Satimage | 0.0176 (5891s) | **0.0981** (2002s) | **0.1699** (4842s) | 0.1058 (2002s) |
| | 2.3 GB | 2.1 GB | 2.3 GB | 2.1 GB |
| Phone | 0.0001 (11s) | **0.1923** (59s) | 0.0023 (11s) | **0.2344** (46s) |
| | 1.0 GB | 1.7 GB | 1.0 GB | 1.5 GB |
| Watch | 0.0000 (973s) | **0.1400** (5230s) | 0.0025 (1017s) | **0.1851** (5371s) |
| | 1.4 GB | 8.7 GB | 1.4 GB | 7.6 GB |
| Still | 0.7900 (12s) | **0.7902** (81s) | **0.8370** (15s) | 0.8368 (117s) |
| | 1.4 GB | 1.6 GB | 1.4 GB | 1.5 GB |

Figure 6: *SNG-DBSCAN against DBSCAN++[25] on large datasets tuned over $\epsilon$.*

| | DBSCAN++ ARI | SNG ARI | DBSCAN++ AMI | SNG AMI |
|---|---|---|---|---|
| Wearable | 0.2517 | **0.3263** | 0.4366 | **0.5003** |
| Iris | **0.6844** | 0.5687 | **0.7391** | 0.7316 |
| LIBRAS | **0.1747** | 0.0904 | **0.3709** | 0.2985 |
| Page Blocks | 0.0727 | **0.1137** | 0.0586 | **0.0760** |
| kc2 | 0.3621 | **0.3747** | 0.1780 | **0.1792** |
| Faces | **0.0441** | 0.0424 | **0.2912** | 0.2749 |
| Ozone | **0.0627** | 0.0552 | 0.1065 | **0.1444** |
| Bank | **0.2599** | 0.2245 | 0.0874 | **0.0875** |
| Ionosphere | 0.1986 | **0.6359** | 0.2153 | **0.5615** |
| Mozilla | 0.1213 | **0.2791** | 0.1589 | **0.1806** |
| Tokyo | 0.4180 | **0.4467** | 0.2793 | **0.3147** |
| Vehicle | **0.1071** | 0.0971 | **0.1837** | 0.1807 |

Figure 7: *Performance comparison of SNG-DBSCAN and DBSCAN++ (with uniform sampling) with $\epsilon$ tuned appropriately and sampling rate tuned over grid [0.1,0.2,..,0.9] to maximize ARI and AMI clustering scores.*

In Figure 7, we compare DBSCAN++ and SNG-DBSCAN on the smaller datasets under optimal tuning of the sampling rate as well as the other hyperparameters. We find that under optimal tuning, these algorithms have competitive performance against each other.

## 6  Conclusion

Density clustering has had a profound impact on machine learning and data mining; however, it can be very expensive to run on large datasets. We showed that the simple idea of subsampling the neighborhood graph leads to a procedure, which we call SNG-DBSCAN that runs in $O(n \log n)$ time and comes with statistical consistency guarantees. We showed empirically on a variety of real datasets that SNG-DBSCAN can offer a tremendous savings in computational resources while maintaining clustering quality. Future research directions include using adaptive instead of uniform sampling of edges, combining SNG-DBSCAN with other speedup techniques, and paralellizing the procedure for even faster computation.

## Broader Impact

As stated in the introduction, DBSCAN has a wide range of applications within machine learning and data mining. Our contribution is a more efficient variant of DBSCAN. The potential impact of SNG-DBSCAN lies in considerable savings in computational resources and further applications of density clustering which weren't possible before due to scalability constraints.

## Acknowledgments and Disclosure of Funding

Part of the work was done when Jennifer Jang was employed at Uber. Jennifer also received unemployment benefits from the New York State Department of Labor during the time she worked on this paper.

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

# A Proofs

Let $\mathcal{P}_n$ be the empirical distribution w.r.t. $X_{[n]}$. We need the following result giving uniform guarantees on the masses of empirical balls with respect to the mass of true balls w.r.t. $\mathcal{P}$.

**Lemma 4** (Chaudhuri and Dasgupta [5]). *Pick $0 < \delta < 1$. Then with probability at least $1 - \delta$, for every ball $B \subset \mathbb{R}^D$ and $k \geqslant D \log n$, we have*

$$\mathcal{P}(B) \geqslant C_{\delta,n} \frac{\sqrt{D \log n}}{n} \Rightarrow \mathcal{P}_n(B) > 0$$

$$\mathcal{P}(B) \geqslant \frac{k}{n} + C_{\delta,n} \frac{\sqrt{k}}{n} \Rightarrow \mathcal{P}_n(B) \geqslant \frac{k}{n}$$

$$\mathcal{P}(B) \leqslant \frac{k}{n} - C_{\delta,n} \frac{\sqrt{k}}{n} \Rightarrow \mathcal{P}_n(B) < \frac{k}{n},$$

*where $C_{\delta,n} := 16 \log(2/\delta) \sqrt{D \log n}$.*

*Proof of Lemma 1.* Suppose that $S$ is a proper subset of the nodes in $G_{n,\epsilon}(i)$. Let $S^C = (X_{[n]} \cap C_i) \backslash S$ be the complement of $S$ in $G_{n,\epsilon}(i)$. We lower bound the cut of $S$ and $S^C$ in $G_{n,\epsilon}(i)$.

Let $r_0 := \left( \frac{16 \log(2/\delta) \cdot D \log n}{\lambda_C \cdot \rho \cdot n} \right)^{1/D}$. Then for any $x \in C_i$, we have:

$$\mathcal{P}(B(x, r_0)) \geqslant \lambda_C \cdot \text{Volume}(B(x, r_0) \cap C_i) \geqslant \lambda_C \cdot \rho \cdot v_D \cdot r_0^D = C_{\delta,n} \frac{\sqrt{D \log n}}{n}.$$

Thus, by Lemma 4, there exists a sample point in $B(x, r_0)$. It follows that $\{B(x, r_0) : x \in S \cup S^C\}$ forms a cover of $C_i$ (i.e $C_i \subseteq \bigcup \{B(x, r_0) : x \in S \cup S^C\}$). Since $C_i$ is connected, then there exists $x \in S$ and $x' \in S^C$ such that $B(x, r_0)$ and $B(x', r_0)$ intersect and thus $|x - x'| \leqslant 2 \cdot r_0$.

The cut of $S$ and $S^C$ in $G_{n,\epsilon}(i)$ thus contains all edges from $x$ to $S^C$ and from $x'$ to $S$, which is at least the number of nodes in $B(x, \epsilon) \cap B(x', \epsilon)$. We have

$$\text{Cut}(S, S^C) \geqslant |B(x, \epsilon) \cap B(x', \epsilon) \cap X_{[n]}| \geqslant |B(x, \epsilon - 2r_0) \cap X_{[n]}| = n \cdot \mathcal{P}_n(B(x, \epsilon - 2r_0)).$$

We have

$$\mathcal{P}(B(x, \epsilon - 2r_0)) \geqslant \lambda_C \cdot \rho \cdot v_D \cdot (\epsilon - 2r_0)^D \geqslant \lambda_C \cdot \rho \cdot v_D \cdot \epsilon^D \cdot \left( 1 - \frac{2Dr_0}{\epsilon} \right) \geqslant \frac{1}{2} \lambda_C \cdot \rho \cdot v_D \cdot \epsilon^D,$$

which holds for $n$ sufficiently large as in the statement of the Lemma for some $C_{D,p}$. By Lemma 4, we have

$$\mathcal{P}_n(B(x, \epsilon - 2r_0)) \geqslant \frac{1}{4} \cdot \lambda_C \cdot \rho \cdot v_D \cdot \epsilon^D,$$

which also holds for $n$ sufficiently large as in the statement of the Lemma for some $C_{D,p}$. The result follows. $\qquad\square$

*Proof of Lemma 2.* Let $N_{n,s}(x, \epsilon)$ be the neighbors of $x$ in the sampled $\epsilon$-neighborhood graph. Suppose that $x \in C_i$ for some $i \in [\ell]$. Then by Hoeffding's inequality, we have

$$P \left( \frac{|N_{n,s}(x, \epsilon)|}{sn} \leqslant \mathcal{P}(B(x, \epsilon)) - \sqrt{\frac{\log(n) + \log(1/\delta)}{2sn}} \right) \leqslant \frac{\delta}{n}.$$

Thus, with probability at least $1 - \delta/n$, we have

$$|N_{n,s}(x, \epsilon)| \geqslant sn \cdot \mathcal{P}(B(x, \epsilon)) - \sqrt{\frac{1}{2} (\log(n) + \log(1/\delta)) \cdot sn}$$

$$\geqslant sn\rho \cdot v_D \cdot \epsilon^D \cdot \lambda_C - \sqrt{\frac{1}{2} (\log(n) + \log(1/\delta)) \cdot sn}.$$

Now, suppose that $x \in N$. Then by Hoeffding's inequality, we have

$$P\left(\frac{|N_{n,s}(x,\epsilon)|}{sn} \geq \mathcal{P}(B(x,\epsilon)) + \sqrt{\frac{\log(n) + \log(1/\delta)}{2sn}}\right) \leq \frac{\delta}{n}.$$

Thus, with probability at least $1 - \delta/n$, we have

$$|N_{n,s}(x,\epsilon)| \leq sn \cdot \mathcal{P}(B(x,\epsilon)) + \sqrt{\frac{1}{2}(\log(n) + \log(1/\delta)) \cdot sn}$$

$$\leq sn \cdot v_D \cdot \epsilon^D \cdot \lambda_N + \sqrt{\frac{1}{2}(\log(n) + \log(1/\delta)) \cdot sn}.$$

Hence, the following holds uniformly with probability at least $1 - \delta$. When $x \in C_i$ for some $i$, we have

$$\frac{|N_{n,s}(x,\epsilon)|}{sn} \geq \rho \cdot \lambda_C \cdot v_D \cdot \epsilon^D - \sqrt{\frac{\log(n) + \log(1/\delta)}{2 \cdot sn}},$$

and otherwise we have

$$\frac{|N_{n,s}(x,\epsilon)|}{sn} \leq \lambda_N \cdot v_D \cdot \epsilon^D + \sqrt{\frac{\log(n) + \log(1/\delta)}{2 \cdot sn}}.$$

The result follows because minPts is the threshold on $\frac{|N_{n,s}(x,\epsilon)|}{sn}$ that decides whether a point is a core-point. There are no border points because $\epsilon < R_S$ and there are no points within $R_S$ of $C_i$ for $i \in [\ell]$. □

*Proof of Lemma 3.* Follows from Theorem 2.1 of [28]. □

*Proof of Theorem 2.* There are two quantities to bound: (i) $\max_{x \in \widehat{L_f(\lambda)}} d(x, L_f(\lambda))$, and (ii) $\sup_{x \in L_f(\lambda)} d(x, \widehat{L_f(\lambda)})$. We start with (i). Define

$$\widehat{f}(x) := \frac{1}{v_D \cdot \epsilon^D} \cdot \frac{\deg_G(x)}{s \cdot n}, \qquad f_\epsilon(x) := \frac{1}{v_D \cdot \epsilon^D} \cdot \int_{x' \in B(x,\epsilon)} f(x')dx',$$

where $\deg_G$ denotes the degree of a node in the subsampled $\epsilon$-neighborhood graph $G$ as per Algorithm 1 and $f_\epsilon(x)$ can be viewed as the density $f$ smoothed with a uniform kernel with radius $\epsilon$. Then, we have for all $x \in X_{[n]}$, the probability that an edge incident to $x$ appearing in $G$ has probability $v_D \cdot \epsilon^D \cdot f_\epsilon(x)$. Thus we have by Hoeffding's inequality:

$$\mathbb{P}(|\widehat{f}(x) - f_\epsilon(x)| \geq \eta) \leq 1 - 2\exp(-\eta^2 \cdot s \cdot n).$$

Setting

$$\eta = \sqrt{\frac{\log(4n) + \log(1/\delta)}{s \cdot n}},$$

we have that by union bound, with probability at least $1 - \delta/2$ uniformly in $x \in X_{[n]}$:

$$|\widehat{f}(x) - f_\epsilon(x)| \leq \sqrt{\frac{\log(4n) + \log(1/\delta)}{s \cdot n}}.$$

Now, for $r > 0$ to be specified later, we have that if $x \in X_{[n]}$ such that $d(x, L_f(\lambda)) \geq r$, then by Assumption 2 for $n$ sufficiently large depending on $f$, $\epsilon$, and MinPts:

$$f(x) \leq \lambda - \check{C}r^\beta \Rightarrow f_\epsilon(x) \leq \lambda - \check{C}(r - \epsilon)^\beta.$$

Therefore,

$$\widehat{f}(x) \leq \lambda - \check{C}(r - \epsilon)^\beta + \sqrt{\frac{\log(4n) + \log(1/\delta)}{s \cdot n}}.$$

Hence, if the following holds:

$$\lambda - \check{C}(r - \epsilon)^{\beta} + \sqrt{\frac{\log(4n) + \log(1/\delta)}{s \cdot n}} < \frac{\text{MinPts}}{v_D \cdot \epsilon^D \cdot s \cdot n},$$

then $x$ is not a core point and hence $\max_{x \in \widehat{L_f(\lambda)}} d(x, L_f(\lambda)) \leqslant r$. This holds by choosing $r$ to be the bound specified in the theorem statement for some $C$ depending on $\check{C}, \widehat{C}, \beta$ by generalized mean inequality. Now we move onto the other direction. We first show that each $x \in L_f(\lambda) \cap X_{[n]}$ is a core point. Indeed we have

$$\widehat{f}(x) \geqslant f_\epsilon(x) - \sqrt{\frac{\log(4n) + \log(1/\delta)}{s \cdot n}} \geqslant \inf_{x' \in B(x,\epsilon)} f(x') - \sqrt{\frac{\log(4n) + \log(1/\delta)}{s \cdot n}}$$

$$\geqslant \lambda - \widehat{C}\epsilon^{\beta} - \sqrt{\frac{\log(4n) + \log(1/\delta)}{s \cdot n}} = \frac{\text{MinPts}}{v_D \cdot \epsilon^D \cdot s \cdot n}.$$

Therefore, each $x \in L_f(\lambda) \cap X_{[n]}$ is a core point. Finally, it suffices to bound the distance from any $x \in L_f(\lambda)$ to some point in $L_f(\lambda) \cap X_{[n]}$. We have for all $x \in L_f(\lambda)$ that

$$\mathcal{P}(B(x,\epsilon)) = v_D \cdot \epsilon^D \cdot f_\epsilon(x) \geqslant v_D \cdot \epsilon^D \cdot (\lambda - \widehat{C}\epsilon^{\beta}) \geqslant \frac{16D \cdot \log(4/\delta) \cdot \log n}{n},$$

for $n$ sufficiently large. Then we have by Lemma 4 that with probability at least $1 - \delta/2$, $\sup_{x \in L_f(\lambda)} d(x, \widehat{L_f(\lambda)}) \leqslant \epsilon$, as desired. □

# B   Additional Experiments

| | $n$ | $D$ | $c$ | $s$ | MinPts | Range of $\epsilon$ |
|---|---|---|---|---|---|---|
| Australian | 1,000,000 | 14 | 2 | 0.001 | 10 | [0.2, 2.5) |
| Still | 949,983 | 3 | 6 | 0.001 | 10 | [0.1, 13.5) |
| Watch | 3,205,431 | 3 | 7 | 0.01 | 10 | [0.001, 0.05) |
| Satimage | 1,000,000 | 36 | 6 | 0.02 | 10 | [1, 6) |
| Phone | 1,000,000 | 3 | 7 | 0.001 | 10 | [0.001, 0.06) |
| Wearable [49] | 165,633 | 16 | 5 | 0.01 | 10 | [1, 80) |
| Iris | 150 | 4 | 3 | 0.3 | 10 | [0.1, 2.2) |
| LIBRAS | 360 | 90 | 15 | 0.3 | 10 | [0.5, 1.6) |
| Page Blocks | 5,473 | 10 | 5 | 0.1 | 10 | [1, 10,000) |
| kc2 [42] | 522 | 21 | 2 | 0.3 | 10 | [50, 7,000) |
| Faces | 400 | 4,096 | 40 | 0.3 | 10 | [6, 10) |
| Ozone | 330 | 9 | 35 | 0.3 | 10 | [100, 800) |
| Bank | 8,192 | 32 | 10 | 0.01 | 10 | [3.7, 7.4) |
| Ionosphere | 351 | 33 | 2 | 0.3 | 10 | [1, 5.5) |
| Mozilla | 15,545 | 5 | 2 | 0.03 | 2 | [1, 7,000) |
| Tokyo | 959 | 44 | 2 | 0.1 | 2 | [10K, 1,802K) |
| Vehicle [43] | 846 | 18 | 4 | 0.3 | 10 | [10, 40) |

Figure 8: *Summary of all datasets used.* We give the datasets used in the main text along with eight additional real datasets. For each dataset, we give $n$ (dataset size), $D$ (number of features), $c$ (number of clusters), and $s$ (sampling rate). We tuned each dataset over 10 equally-spaced $\epsilon$ values in the range given. We chose $s$ for each dataset from a small set of values roughly depending on the size of the dataset. In theory, $s$ can be chosen as $O(\log n/n)$, but there is a constant multiplier that is data-dependent, and it was much simpler to tune $s$ over a small grid, which is described in the captions of the experiment results.

Figure 9: **Small datasets**. We show the performance of SNG-DBSCAN and DBSCAN on 12 smaller datasets. Runtimes and scores for both SNG-DBSCAN and DBSCAN are averaged over 10 runs, and 95% confidence intervals (standard errors) are shown. We ran these experiments on a local machine with 16 GB RAM and a 2.8 GHz Quad-core Intel Core i7 processor. We used $s \in \{0.01, 0.03, 0.1, 0.3\}$ depending on the size of the dataset, across a wide range of epsilon, and with MinPts $= 10$ for most datasets, other than Mozilla and Tokyo where we used MinPts $= 2$.

Figure 10: **Sampling rates**. We plot the performance of SNG-DBSCAN across different sampling rates $s$ to be better understand the effect of $s$ on the clustering quality. We make comparisons to the best performance of DBSCAN shown as the dotted vertical lines for each score. SNG-DBSCAN is run on the same epsilon as the DBSCAN benchmark for that dataset. SNG-DBSCAN values are averaged over 10 runs, and 95% confidence intervals (standard errors) are shown. We see that SNG-DBSCAN converges quickly to comparative or better clusters even at low sampling rates suggesting that often-times a small $s$ suffices, highlighting both the stability and scalabiltiy of SNG-DBSCAN.

|  | Base ARI | SNG ARI | Base AMI | SNG AMI |
|---|---|---|---|---|
| Wearable | 0.2626 (55s) | **0.3064** (8.2s) | **0.4788** (26s) | 0.4720 (6.6s) |
| Iris | **0.5681** (<0.01s) | **0.5681** (<0.01s) | **0.7316** (<0.01s) | **0.7316** (<0.01s) |
| LIBRAS | 0.0713 (0.03s) | **0.0903** (<0.01s) | 0.2711 (0.02s) | **0.3178** (<0.01s) |
| Page Blocks | 0.1118 (0.38s) | **0.1134** (0.06s) | **0.0742** (0.49s) | 0.0739 (0.06s) |
| kc2 | 0.3729 (<0.01s) | **0.3733** (<0.01s) | **0.1772** (<0.01s) | 0.1671 (<0.01s) |
| Faces | 0.0345 (1.7s) | **0.0409** (0.16s) | 0.2399 (1.7s) | **0.2781** (0.15s) |
| Ozone | 0.0391 (<0.01s) | **0.0494** (<0.01s) | 0.1214 (<0.01s) | **0.1278** (<0.01s) |
| Bank | 0.1948 (3.4s) | **0.2265** (0.03s) | 0.0721 (3.3s) | **0.0858** (0.03s) |
| Ionosphere | 0.6243 (0.01s) | **0.6289** (<0.01s) | **0.5606** (0.01s) | 0.5437 (<0.01s) |
| Mozilla | 0.1943 (0.08s) | **0.2642** (0.05s) | 0.1452 (0.09s) | **0.1558** (0.05s) |
| Tokyo | **0.4398** (0.02s) | 0.4379 (<0.01s) | 0.2872 (0.02s) | **0.3053** (<0.01s) |
| Vehicle | **0.0905** (0.01s) | 0.0845 (<0.01s) | 0.1643 (<0.01s) | **0.1653** (<0.01s) |

Figure 11: *Clustering performance for additional datasets.* Best Adjusted Rand Index and Adjusted Mutual Information scores for both DBSCAN and SNG-DBSCAN under best hyperparameter tuning are given with associated runtimes in parentheses.