[Reviews · NeurIPS 2020]

Review 1

Summary and Contributions: Authors provide a new version of DBSCAN that is optimized for time complexity by sub-sampling edges of the epsilon-neighboring graph on which clustering partition is computed. Proposed time complexity is O(sn**2) where: -n is the number of data points -s is the sampling rate of the edges

Strengths: - Authors support the approach with theoretical guarantees: under some assumptions, s ~ (log n) / n is proven to be enough for ensuring statistical recovery of the clustering partition. Demonstrations are well conducted: interpretation of each assumption makes it easy to follow. - Empirical evaluation show that competitive performance (quality of the clustering and runtime) for big and even small datasets.

Weaknesses: - Empirical evaluation is a bit too weak for me. The comparison with DBSCAN++ is not fair. Even if DBSCAN++ and approach proposed by the authors come up with the same bound for runtime complexity O(sn**2) with n the number of nodes and s the sampling rate, this is definitely not fair to use the same sampling rate for edges (authors' approach) and DBSCAN++. Hence could the authors provide experiments for which sampling rate is optimized for both competitors? - Authors DBSCAN++ show cases where DBSCAN++ is even providing better clustering partition than for DBSCAN. Could the authors perform a similar analysis here with SNG-DBSCAN? - DBSCAN++ is also applied to outlier detection. How is performing SNG-DBSCAN compared to DBSCAN++ in this context? - Which parameters are used with DBSCAN for time and space consumption comparison? - Would it be possible to have empirical results on datasets for which DBSCAN is actually working? It seems that DBSCAN is performing well only for Still. Hence DBSCAN and SNG-DBSCAN are equally performing bad on the remaining sets...

Correctness: My main concern is the fact that SNG-DBSCAN and DBSCAN++ are empirically compared by using the same sampling rate while for SNG-DBSCAN we sample edges and for DBSCAN++ we sample nodes. Hence, SNG-DBSCAN has naturally an advantage over DBSCAN regarding clustering accuracy which is not fair. Authors should present the results by tuning the sample rate for both methods.

Clarity: The paper is undoubtedly well written and easy to follow. Each theoretical result is well presented before the proof.

Relation to Prior Work: Yes. Main previous contribution is mainly DBSCAN ++ for the last optimization of DBSCAN. SNG-DBSCAN is about sampling edges rather than nodes.

Reproducibility: Yes

Additional Feedback: As it is about optimization of a well-established algorithms and execution times are compared with scikitlearn implementation, it could be very valuable the authors provide actually access to the code. === AFTER AUTHOR FEEDBACK === My main concern about the mise-use of the same sampling rate for edges vs nodes has been correctly addressed by the authors and I trust them to do the appropriate changes in the revised version of the paper. Hence I upgrade my rating to a weak accept.


Review 2

Summary and Contributions: The paper proposes a new approach for speeding up the popular DBSCAN algorithm for density based clustering. The speed up is based on sampling edges in the e-neighbourhood graph, where previous sampling based speed ups arise from sampling from the data instead. The authors provide a theoretical guarantee for the correct recovery of the level set of a density with high probability, provided assumptions on the density hold. The experiments provided illustrate potential for practical relevance when clustering large data sets

Strengths: The paper is concisely and clearly written. The proposed idea is simple and intuitive, but appears to be novel. It also seems to work very well practically. DBSCAN is a highly influential clustering algorithm, and improvements thereon have potential for substantial impact.

Weaknesses: The paper includes a theoretical guarantee for the algorithm which ensures correct recovery of the level set components of the underlying density with high probability, however the assumptions required for this to hold seem to be very restrictive. Aside from being far more restrictive than in studies such as those in [1] and [2] which also deal with the estimation of level sets and their components, the assumptions are not very natural. In particular, as far as it seems, the density must be discontinuous at the boundaries of the level set and the noise region must be separated from the clusters by a regions of zero probability. While this is not stated explicilty, it can be inferred from the assumptions and the conclusion in Theorem 1 which states that the exact clusters are recovered, without the potential for points just beyond the boundary of the level set, which with a continuous density would have density arbitrarily close to the height of the density inside the level set. References: [1] Steinwart, Ingo. "Fully adaptive density-based clustering." The Annals of Statistics 43.5 (2015): 2132-2167. [2] Hofmeyr, David P. "Connecting Spectral Clustering to Maximum Margins and Level Sets." Journal of Machine Learning Research 21.18 (2020): 1-35.

Correctness: The results and claims appear to be correct (given the assumptions), and the empirical methodology is straightforward but compelling.

Clarity: See above in the strengths section

Relation to Prior Work: Given the limited space the authors have done well to mention a large number of related prior works. Under some circumstances the lack of detail would be a concern, but given the relative simplicity of the DBSCAN algorithm and sampling based speed ups, I think the coverage is sufficient.

Reproducibility: Yes

Additional Feedback: EDIT: After reading the authors' response and discussing with other reviewers, I am satisfied that the authors can appropriately address my concerns. Additional to the concern mentioned above regarding the assumptions on the underlying density, unless I am mistaken the conditions on \rho are also very restrictive and could even have been written incorrectly? In particular, even in the ideal scenario of a very "nicely" shaped cluster, a point x near the boundary of the cluster C would result in Vol(B(x, r) ^ C) \approx Vol(B(x, r)) / 2, meaning that \rho can never be greater than 1/2. Applying this in the first point in Assumption 1 suggests that we can't use this result if any of the clusters has a density only \approx twice that of the height of the noise density. I would be surprised if the assumptions cannot be relaxed to a large degree to ensure the same or a very similar result will hold under far more general and natural conditions. I would be very willing to recommend acceptance if the authors can address my concerns regarding the relevance of the theory, as the empirical performance of the method is very persuasive. In addition, I found a few typos/curiosities: 1. kd-trees (line 113) and KD-trees (line 40) are both used 2. line 123 repetition of the word "degree" in the parentheses 3. In the statement of Lemma 1 the first instance of C_{D, p} also has a \delta in the subscript


Review 3

Summary and Contributions: 1. DBSCAN is a popular density-based clustering algorithm. This paper proposes a faster variant of DBSCAN called SNG-DBSCAN by clustering the subsampled -neighborhood graph instead of computing the entire graph. They show that under some natural assumptions the algorithm provides comparable clustering performance at lower computational costs. The theoretical work depends heavily on statistical as well as graph theory lemmas and theorems. They compare the performance with DBSCAN and DBSCAN++ for some datasets.

Strengths: The paper converts the epsilon neighborhood of points as a graph of vertices and the distance between them as edges and applies principles of graph theory as well as statistical sampling theory to prove the claim that the cluster quality is ensured. The paper also relies on previous work of sampling in graph theory and applies it for epsilon neighborhood graph. The paper provides extensive results against two previous techniques, DBSCAN and DBSCAN++. They also provide memory usage metric.

Weaknesses: The already existing work of DBSCAN++ subsamples the vertices of the graph while the proposed SNG-DBSCAN subsamples the edges for better time complexity, as such the idea is not very novel and draws from previous works.

Correctness: The claims and method are correct.

Clarity: The theoretical work in the paper needs to be written more clearly. It is missing out on explaining basic symbols and terminology used and as such makes the work difficult to understand without the supplementary material. The theory written appears vague and would lead one to think that there are several missed points till we dig deeper into supplementary material. The authors should have spent more space on presenting their robust theoretical work rather than providing long explanations of experiments.

Relation to Prior Work: The paper briefly discusses how it differs from previous contributions but fails to draw our attention to the novelty of their approach.

Reproducibility: Yes

Additional Feedback: N/A


Review 4

Summary and Contributions: The paper proposes an edge subsampling approximation to DBSCAN along with theoretical conditions under which it is likely to recover identical results to full DBSCAN, and provides some experimentation over clustering accuracy, time taken, and memory consumed.

Strengths: DBSCAN is widely used clustering algorithm with known scalability problems, so this is a valuable problem to consider. The approach is pleasingly simple, the theoretical analysis provides some explanation of why it should work, and the experimental results confirm the improved scaling performance without degradation of quality or accuracy.

Weaknesses: The connection between the theoretical analysis and the experimental results was not very strongly established: what are the constants hiding in the Big Theta, can we take any guidance for how to set s from this? It was also somewhat surprising to see SNG actually doing better than base DBSCAN in Figure 3, when we should expect identical results under the theory? How do results degrade if the conditions are not met exactly? Without some tighter connection, the theory would seem to serve as a somewhat hand-wavy justification for a very clever and effective implementation trick. This is not without value, but a more direct connection would strengthen the submission.

Correctness: I followed but did not rigorously check the derivations. The experiments seemed reasonable modulo my other questions/comments.

Clarity: The paper is clear overall and in particular the structure of the theoretical analysis was nicely laid out in terms of the necessary assumptions and the resulting conclusion. The plain language parenthetical explanations of the different conditions in L151-158 were helpful, as was the overview of the approach starting on L159.

Relation to Prior Work: As admitted by the authors, the DBSCAN literature is vast. The related work is generally well-contextualized. One question area was around how this work compares to the data-parallel (eg, Map-Reduce) class of modifications: how exactly could those apporaches be easily composed with SNG-DBSCAN? Or, what are the tradeoffs of SNG-DBSCAN versus these? The comparison with DBSCAN++ and conceptual framing as node-vs-edge sampling was useful, although it is not obvious to me that using the same sampling rate s is really an appropriate comparison when one algorithm is sampling nodes while the other is sampling edges?

Reproducibility: Yes

Additional Feedback: It wasn't obvious to me what the "true clusters" meant exactly, in terms of whether we are talking about "the exact results standard DBSCAN would be expected to return" or "the true clustering we would expect based on the underlying density P"? Or is this some intrinsic property of the assumptions set up in terms of C1, ...? Do the sets C1,... need to entirely cover the support of P? Do specific steps of the proof depend on them being compact and connected, or are these just some general "sanity" conditions? Figure 4: I didn't see the standard errors on these charts? In several cases comparisons to other DBSCAN implementations are explained by the fact that the other implementations exploit spatially-aware data structures such as kd trees. Are there any potential future extensions to leverage those in this sampling framework? I have read and considered the author feedback.

[Author Response · NeurIPS 2020]

We thank the reviewers for their thoughtful feedback and for appreciating the simplicity and potentially wide impact of
our results. Due to lack of space, we could only address the major comments, and in this process we add new theoretical
and experimental developments which we will add to the paper if accepted.

**R1**: It seems the main concern is that DBSCAN++ and SNG-DBSCAN are compared using the same sampling rate
which may not be fair as they may not necessarily have the same meaning. The reviewer brings up a good point. To this
end, we provide Figure 1 which shows that SNG-DBSCAN is still competitive when both algorithms are optimized over
both $\epsilon$ and sampling rate. We also note that these procedures may outperform DBSCAN simply because the sampling
adds an additional degree of freedom and can be interpreted as a regularizer [25].

**R2**: The main concern is that the theoretical results have strong assumptions. The reviewer is right. Below, we give level-
set estimation rates for SNG-DBSCAN under more standard and general non-parametric assumptions. The assumptions
are borrowed from other works in level-set estimation (i.e. [26, 44]). Given these results, we can straightforwardly
extended them to obtain clustering results with the same convergence rates (i.e. showing that SNG-DBSCAN recovers
the connected components of the level-set individually), but omit it here due to space.

**R3**: The main concern appears to be the novelty of SNG-DBSCAN relative to DBSCAN++. We emphasize that
although one samples edges and the other samples vertices, there are still considerable differences: they lead to different
theoretical analyses, SNG-DBSCAN appears to perform better, SNG-DBSCAN works for arbitrary distance metrics,
and unlike DBSCAN++, SNG-DBSCAN can be easily used in practice by plugging in a subsampled distance matrix
into scikit-learn's DBSCAN implementation under the precomputed distance setting.

**R5**: The true clusters are the connected components of a particular level-set of the density function. We show that
SNG-DBSCAN recovers these clusters at rates depending on various properties of the density function. The reviewer
is right that since these rates depend on constants that are unknown in practice, they may have little practical use but
nonetheless makes the algorithm a principled approach. We will further clarify these constant factor dependencies.

**Additional Theory**. We show level-set estimation rates for esti-
mating a particular level $\lambda$ (i.e. $L_f(\lambda) := \{x \in \mathcal{X} : f(x) \geq \lambda\}$)
given that hyperparameters of SNG-DBSCAN are set appropriately
depending on density $f$, $s$, $\lambda$ and $n$.

**Assumption 1.** *$f$ is a uniformly continuous density on compact*
*set $\mathcal{X} \subseteq \mathbb{R}^D$. There exists $\beta, \check{C}, \hat{C}, r_c > 0$ such that the following*
*holds for all $x \in B(L_f(\lambda), r_c) \backslash L_f(\lambda)$: $\check{C} \cdot d(x, L_f(\lambda))^\beta \leq \lambda -$*
*$f(x) \leq \hat{C} \cdot d(x, L_f(\lambda))^\beta$, where $d(x, A) := \inf_{x' \in A} |x - x'|$,*
*$B(C, r) := \{x \in \mathcal{X} : d(x, C) \leq r\}$.*

where $\beta$ can be interpreted as the smoothness and curva-
ture of $f$ around the $\lambda$-level-set boundary of $f$. Define
$C_{\delta,n} = 16 \log(2/\delta)\sqrt{\log n}$, $\epsilon = (\text{minPts}/(sn \cdot v_D \cdot (\lambda -$
$\lambda \cdot C_{\delta,n}^2/\sqrt{\text{minPts}})))^{1/D}$, and minPts satisfies $C_l \cdot (\log n)^2 \leq$
$\text{minPts} \leq C_u \cdot (\log n)^{\frac{2D}{2+D}} \cdot n^{2\beta/(2\beta+D)}$ where $C_l$ and $C_u$ are
positive constants depending on $\delta, f$. Then, the following holds
where $d_{\text{Haus}}$ is Hausdorff distance:

**Theorem 1.** *Suppose Assumption 1 holds along with the parame-*
*ter settings of the above. There exists $C, C_l, C_u > 0$ depending on*
*$f, \delta$ such that the following holds with probability at least $1 - \delta$. Let*
*$\widehat{L_f(\lambda)}$ be the union of all the clusters returned by SNG-DBSCAN:*

| | | DBSCAN | DBSCAN++ | SNG |
|---|---|---|---|---|
| Page | | 0.1118 | 0.0727 | **0.1137** |
| Blocks | | 0.0742 | 0.0586 | **0.0760** |
| kc2 | | 0.3729 | 0.3621 | **0.3747** |
| | | 0.1772 | 0.1780 | **0.1792** |
| Ozone | | 0.0391 | **0.0627** | 0.0552 |
| | | 0.1214 | 0.1065 | **0.1444** |
| Bank | | 0.1948 | **0.2599** | 0.2245 |
| | | 0.0721 | 0.0874 | **0.0875** |
| Ionosphere | | 0.6243 | 0.1986 | **0.6359** |
| | | 0.5606 | 0.2153 | **0.5615** |
| Mozilla | | 0.1943 | 0.1213 | **0.2791** |
| | | 0.1452 | 0.1589 | **0.1806** |
| Tokyo | | 0.4204 | 0.4180 | **0.4467** |
| | | 0.2830 | 0.2793 | **0.3147** |

Figure 1: *DBSCAN tuned over $\epsilon$ and SNG-DBSCAN and DBSCAN++ (which uniformly samples the nodes) tuned over $\epsilon$ (same grid as in paper for each dataset) and sampling rate (over grid [0.1,0.2,..,0.9]) to maximize ARI and AMI clustering scores. Only some datasets shown.*

$$d_{Haus}(\widehat{L_f(\lambda)}, L_f(\lambda)) \leq C \cdot \left( C_{\delta,n}^{2/\beta} \cdot minPts^{-1/2\beta} + C_{\delta,n}^{1/D} \cdot \left( \frac{\sqrt{\log sn}}{sn} \right)^{1/D} \right) \rightarrow_{sn/\log(n), n \to \infty} 0.$$

*Proof Sketch.* There are two quantities to bound: (i) $\max_{x \in \widehat{L_f(\lambda)}} d(x, L_f(\lambda))$, and (ii) $\sup_{x \in L_f(\lambda)} d(x, \widehat{L_f(\lambda)})$. The
bound for (i) follows by standard uniform kernel density (KDE) estimation bounds with uniform kernel (i.e. [26]) based
on the $sn$ samples where the first term in the rate is due to the bias of the smoothing w.r.t. $\epsilon$ and the variance term comes
from sampling at a rate of $s$ for each estimate. We now turn to the other direction and bound $\sup_{x \in L_f(\lambda)} d(x, \widehat{L_f(\lambda)})$.
Let $x \in L_f(\lambda)$. Define $r_0 := ((2C_{\delta,n}\sqrt{D \log sn})/(snv_D \cdot \lambda))^{1/D}$. Using standard concentration inequalities, we show
that $B(x, r_0)$ contains at least $1/s$ samples and by standard density estimation guarantees, at least one of them will
have sufficiently high KDE with uniform kernel and bandwidth $\epsilon$ leading to the conclusion that its $\epsilon$-ball contains at
least MinPts edges after subsampling at a rate of $1/s$. Thus, $\sup_{x \in L_f(\lambda)} d(x, \widehat{L_f(\lambda)}) \leq r_0$. $\square$

[Meta-Review · NeurIPS 2020]

During the discussion, reviewers agree with the strength of this paper about a simple but novel idea, which is also theoretically supported. The most crucial concern raised by the reviewers is the miss-use of sampling rate in experiments, while it is successfully addressed in author response and reviewers acknowledge this. Hence I recommend acceptance of the paper.